# Data-Independent Acquisition Enables Robust Quantification of 400 Proteins in Non-Depleted Canine Plasma

**DOI:** 10.3390/proteomes10010009

**Published:** 2022-02-28

**Authors:** Halley Gora Ravuri, Zainab Noor, Paul C. Mills, Nana Satake, Pawel Sadowski

**Affiliations:** 1School of Veterinary Science, The University of Queensland, Gatton, QLD 4343, Australia; h.ravuri@uq.edu.au (H.G.R.); p.mills@uq.edu.au (P.C.M.); 2ProCan^®^, Children’s Medical Research Institute, Faculty of Medicine and Health, The University of Sydney, Westmead, NSW 2145, Australia; znoor@cmri.org.au; 3Central Analytical Research Facility, Queensland University of Technology, Brisbane, QLD 4000, Australia

**Keywords:** dog, plasma proteomics, FASP, Data Independent Acquisition, SWATH

## Abstract

Mass spectrometry-based plasma proteomics offers a major advance for biomarker discovery in the veterinary field, which has traditionally been limited to quantification of a small number of proteins using biochemical assays. The development of foundational data and tools related to sequential window acquisition of all theoretical mass spectra (SWATH)-mass spectrometry has allowed for quantitative profiling of a significant number of plasma proteins in humans and several animal species. Enabling SWATH in dogs enhances human biomedical research as a model species, and significantly improves diagnostic and disease monitoring capability. In this study, a comprehensive peptide spectral library specific to canine plasma proteome was developed and evaluated using SWATH for protein quantification in non-depleted dog plasma. Specifically, plasma samples were subjected to various orthogonal fractionation and digestion techniques, and peptide fragmentation data corresponding to over 420 proteins was collected. Subsequently, a SWATH-based assay was introduced that leveraged the developed resource and that enabled reproducible quantification of 400 proteins in non-depleted plasma samples corresponding to various disease conditions. The ability to profile the abundance of such a significant number of plasma proteins using a single method in dogs has the potential to accelerate biomarker discovery studies in this species.

## 1. Introduction

Dogs are proving to be a useful model for many human diseases, such as osteosarcoma [1], prostate cancer [2], and lymphoma [3], and for developing therapeutic protocols during clinical trials [4]. Dogs are also useful translational models for epilepsy [5], autism [6], and the investigation of various zoonotic diseases [7]. Moreover, as companion animals, their health status and well-being are of utmost importance to their owners. Blood plasma is routinely collected from dogs and analyzed to assess health; however, surprisingly little assays are available for quantification of proteins in this species. Current veterinary diagnostic approaches are limited to targeting highly abundant proteins, such as albumin and globulins, using biochemical assays. Several major acute phase proteins (APPs) and a few cytokines can be measured using traditional protein quantitation methods, including enzyme-linked immunosorbent assays (ELISA), and radioimmune assays (RIA), but they were developed for other species and assume cross-reactivity with specific dog proteins [8]. Importantly, many oxidative markers and tissue leakage proteins are not routinely assessed due to the unavailability of relevant diagnostic kits. 

Mass spectrometry offers a unique opportunity to quantify hundreds to thousands of proteins in a single assay; however, highly abundant proteins must be depleted to avoid interference with the detection of lower abundant proteins when protein quantitation is conducted using data-dependent acquisition (DDA)-based approaches [9]. The data-dependent acquisition (DDA) approach used to be a standard data acquisition mode to identify proteins using LC-MS/MS in various biomarker discovery experiments. In this method, the most abundant ions (MS) entering the mass spectrometer are selected (in a small isolation window, with <1 da) for MS/MS fragmentation. Each of these fragmented MS/MS scans is analyzed against proteome database search engines for the identification of proteins in samples [10]. Despite its successful application to biomarker discovery studies [11], the DDA technique is inherently biased with the selection of abundant precursor ions, which compromises the reproducible detection of smaller and less abundant peptides, as it also does with proteins [12,13]. 

The introduction of sequential window acquisition of all theoretical mass spectra (SWATH), a data-independent acquisition (DIA) approach available on selected mass spectrometry platforms [14], has overcome the limitation of abundance bias and has proven useful in quantifying proteins in non-depleted plasma samples from humans and laboratory animals [15,16]. In this acquisition mode, all peptides entering a mass spectrometer (irrespective of their abundance) will be fragmented (in consecutive isolation windows ranging between 10–50 da) within a given mass window (400–1200 *m*/*z*) [17], which enables reproducible quantitation of peptides from complex sample mixtures. However, SWATH data analysis relies on the availability of a highly curated DDA based reference spectral library, which should be well-annotated with a genome and contains information about peptide sequences, their chromatographic elution times, and characteristic product ions [18], and a significant community effort has been made to develop relevant resources for several species [19,20]. With the availability of spectral libraries in humans [21], SWATH is now extensively used to explore the pathophysiology of diseases in humans [22,23]. Recently, very few studies have reported the application of SWATH for the quantitation of proteins from plasma in dogs [24]; cattle [25] and sheep [26]. Most of these studies first developed an in-house DDA spectral library for enabling SWATH quantitation. Similarly, this study also utilized DDA mass spectrometry to generate a peptide spectral library in combination with several orthogonal fractionation and digestion approaches to identify the highest number of proteins over a previously published SWATH study [24]. In addition, we also proved that, when combined with a microflow liquid chromatography and a variable windows variant of SWATH, the library enabled development of a quantitative assay for robust monitoring of 400 proteins in clinically relevant and non-depleted plasma samples. We will share this development with the scientific community via PeptideAtlas portal [27] to underpin future studies in dogs. 

## 2. Materials and Methods

### 2.1. Sample Collection

A total of 34 dog blood plasma samples (detailed information of each sample were provided as Appendix A) were obtained from archived samples stored in the Veterinary Diagnostic Laboratory in the School of Veterinary Science at The University of Queensland. The samples had been collected from clinically healthy subjects and animals with different clinical conditions (Table 1). This study has been approved by The University of Queensland Animal Ethics Committee (ANRFA/SVS/541/18). 

### 2.2. Sample Preparation 

All samples were acetone precipitated and subjected to various digestion and fractionation/enrichment techniques depending on the purpose of analysis, qualitative (Table 1) or quantitative (Table 2). 

#### 2.2.1. Acetone Precipitation

Samples were precipitated with cold acetone at 1:4 ratio and incubated at −20 °C overnight. After incubation, precipitated samples were centrifuged at 250× *g* for 3 min and the pellet was resuspended in freshly prepared 8 M urea in 25 mM ammonium bicarbonate (AMBIC). The protein concentration was determined using the BCA protein assay kit (Pierce, Thermo Fisher Scientific, San Diego, CA, USA) as per the manufacturer’s instructions. 

#### 2.2.2. In-Solution Digestion

The method followed was established by [28]. In brief, 20 μg of plasma proteins were reduced using dithiothreitol (DDT) (5 mM final concentration) and alkylated using iodoacetamide (IAM) (14 mM final concentration). The reaction was quenched by the addition of dithiothreitol (5 mM final concentration), and calcium chloride (CaCl_2_) was added (10 mM final concentration). Protein digestion was performed adding sequencing grade modified trypsin (Promega, Madison, WI, USA) with 1:50 enzyme: protein ratio and agitated overnight at 37 °C. The next day, the resulting peptides were dried under vacuum and subjected to desalting. 

#### 2.2.3. Filter-Aided Digestion (FASP)

The method followed was established by [29]. Briefly, 20 µg protein was mixed with Sodium dodecyl-sulfate (SDS)-Tris lysis buffer (4% SDS, 100 mM DTT in 100 mM Tris-HCl buffer pH 7.6) and transferred on to a 30 kDa Microcon YM-30 centrifugal filter unit (Millipore, Merck KGaA, Darmstadt, Germany). Next, samples were mixed with DTT-Urea buffer (8 M urea and 25 mM DTT in 100 mM Tris-HCL pH 8.5) and the mixture was agitated for 1 h at room temperature. All the samples were centrifuged at 14,000× *g* for 15 min and mixed with IAM-Urea buffer (8 M urea and 50 mM iodoacetamide in 100 mM Tris-HCL pH 8.5). The samples were centrifuged again followed by two washes with 100 mM AMBIC. Finally, trypsin was added at an enzyme to protein ratio of 1:50 and the samples were agitated overnight at 37 °C. The following day, the peptides were eluted using 100 mM AMBIC into a clean centrifuge tube, dried under vacuum, and subjected to desalting.

#### 2.2.4. ProteoMiner Enrichment 

The ProteoMiner protein enrichment kit enables enrichment of medium- and low-abundance proteins that cannot be detected through traditional methods. For depleting purposes, plasma samples were processed according to the manufacturer’s instructions. Before processing plasma samples, all the spin columns were centrifuged at 1000× *g* for 60 s to remove all the storage material. The column was washed by adding 600 µL Wash buffer, vortexed for 5 min, followed by centrifugation at 1000× *g* for 60 s, and the buffer was discarded. This step was repeated twice, involving washing and discarding the buffer—before plasma samples were processed. A total of 1000 µL of plasma sample (with protein concentration > 50 mg/mL) was added to the column and incubated for 2 h at 30 °C. After incubation, the column was centrifuged at 1000× *g* for 60 s to remove the excess sample. Ligand beads were washed with the wash buffer by rotating for 5 min, followed by centrifugation at 1000× *g* for 60 s. Ligand beads were washed repeatedly three times before the elution process. After a couple of washings with the wash buffer, the column was washed with 600 µL of de-ionized water by rotating for 1 min, followed by centrifugation at 1000× *g* for 60 s. The flow-through was discarded and the elution of the proteins was conducted by adding 100 µL of elution reagent to the column. The column was vortexed for 15 min, followed by centrifugation at 1000× *g* for 60 s. The flowthrough was collected into a labeled collection tube. The elution process was repeated six times, all the elutes were pooled and were fractionated by SDS-PAGE and digested using the in-gel digestion protocol. 

#### 2.2.5. SDS-PAGE Fractionation and In-Gel Digestion

SDS–PAGE fractionation of samples was conducted based on a previously published method [30]. Briefly, 50 µg protein was mixed with a sample buffer containing freshly added DDT (20 mM final concentration) and incubated at 60 °C for 3 min before loading onto the gel. After completing the electrophoresis, the gel was stained with an Aqua stain solution for 5 h and then destained overnight with water. The gel was then cut into 5 mm slices and processed through in-gel digestion. In-gel digestion was performed on individual protein bands using a previously published method [31]. Gel slices were incubated in 50% ACN/100 mM AMBIC at 4 °C for 30 min, followed by incubation with 100% ACN at room temperature. The process was repeated until there was no visible staining. Subsequently, the slices were cut into smaller pieces which were subjected to simultaneous reduction and alkylation using 10 mM Tris(2-carboxyethyl) phosphine hydrochloride (TCEP) and 40 mM chloroacetamide (CAA) in 100 mM AMBIC for 30 min. The gel pieces were then allowed to dry completely under a vacuum. Trypsin at an enzyme: protein ratio of approximately 1:50 based on the band intensity was added and incubated for 30 min at 4 °C. Then 100 mM AMBIC was added, and the samples were incubated at 37 °C overnight on an agitator. The next day, trypsin digestion was stopped by using 4% formic acid. Peptides were extracted from gel pieces by agitating with 50% and 100% ACN and the samples were dried under vacuum and subjected to desalting.

#### 2.2.6. Acetonitrile Precipitation 

A modified method established by [32] was followed with some modifications. Aliquots of raw plasma samples were diluted at a 1:10 ratio with 100 mM ammonium acetate and their pH was adjusted to 3.5; 5; 8; or 9 using either 100% acetic acid or 100 mM ammonium bicarbonate. The samples were then mixed at a 1:1 ratio with 100% acetonitrile and centrifuged at 16,000× *g* for 10 min. Supernatant and pellet fractions from each aliquot were collected separately and dried under vacuum. The dried samples were resuspended in 8 M urea in 25 mM AMBIC. The protein concentration was determined using BCA protein assay as per the manufacturer’s instruction, and the samples were fractionated by SDS PAGE and digested using the in-gel protocol.

#### 2.2.7. Desalting

Trypsin digested proteins were resuspended in 2% ACN in 0.1% TFA and subjected to solid-phase extraction using a SCX membrane disk inserted into a StageTip [33]. Specifically, SCX material was activated using 100% ACN and conditioned with 5% ammonium hydroxide in 80% ACN. SCX membrane equilibration, sample loading, and two washes utilized 2% ACN in 0.1% TFA. Finally, the peptides were eluted using 5% ammonium hydroxide in 80% ACN. Samples were then dried under vacuum and peptides were resuspended using a solution containing 11 standard peptides (iRT Kit from Biognosys) made up in 2% ACN in 0.1% FA and submitted for mass spectrometry analysis.

### 2.3. Mass Spectrometry

#### 2.3.1. Data-Dependent Acquisition (DDA)

For spectral library construction, peptide samples were analyzed on a TripleTOF 5600+ quadrupole time-of-flight mass spectrometer (SCIEX) operated in data-dependent acquisition (DDA) mode. The instrument was equipped with a Nanospray III ion source and coupled to the Eksigent ekspert nanoLC 400 System (Eksigent Technologies, Dublin, CA, USA) [26]. Chromatographic separation of peptides was performed using 150 mm × 75 μm column maintained at 40 °C and packed with reverse phase material (ChromXP C18 3 μm 120 Å) preceded by trapping using 10 mm × 0.3 mm trap cartridge packed with similar material (ChromXP C18CL 5 μm 120 Å). The mobile phase A was water/0.1% FA, mobile phase B—ACN/0.1% FA, and mobile phase C (for trapping)—2% ACN/0.1% FA. Peptides were trapped for 5 min at 5 μL/min followed by an elution across a 90 min run time using mobile phases A and B at a conserved flowrate of 300 nL/min. On DDA mode, a mass spectrometer performed high-resolution TOF-MS scans over 350–1350 *m*/*z* range, followed by up to 40 high sensitivity MS/MS scans of the most abundant peptide ions per cycle over the range of 100–2000 *m*/*z*. Peptides with intensity greater than 150 cps and a charge state of 2–5 were selected for the analysis. The dynamic exclusion duration was adjusted depending on the peak widths. Each TOF-MS scan was performed for 250 ms and the product ion (MS/MS) scan was acquired for 50 ms, resulting in a total cycle time of 2.3 s. The ions were further fragmented in the collision cell, with collision energy spread (CEs) set at 5, and the fragmented peptide ion spectra were saved in wiff format (SCIEX).

#### 2.3.2. Spectral Library Construction

The protein sequence database was assembled in FASTA format downloaded on 20 March 2020 from a repository consisting of 45,351 proteome sequences non-redundant and predicted protein sequences of *Canis lupus familiaris* sourced from UniProtKB/Swiss-Prot (Universal Protein Resource Knowledgebase—http://www.uniprot.org/, accessed on 20 March 2020). The MS/MS spectra were annotated with amino acid sequences imported from canine database using the Paragon Algorithm provided in ProteinPilot v5.0 software (SCIEX) with the following search parameters: Digestion—Trypsin; Instrument—TripleTOF5600; Special Factors—Urea denaturation; Search effort—Thorough ID; ID Focus—Amino acid substitution. The output file in group format was next used to build a spectral library inside Skyline software (Version 21.1.0.278) in blib format according to the published protocol [34]. The false discovery rate at the peptide level was set to 1% during data import.

#### 2.3.3. Data-Independent Acquisition

For quantitative analysis, peptide samples were analyzed as four technical replicates on a TripleTOF 6600 quadrupole time-of-flight mass spectrometer (SCIEX) operated in data-independent acquisition (DIA) mode. For quantitative comparative SWATH analysis, all undepleted plasma samples (healthy, inflammatory, and miscellaneous) were digested using FASP with five technical replicates per condition. All the digested samples were analyzed using a variant of Data Independent Acquisition (variable windows SWATH) on TripleTOF 6600 quadrupole time-of-flight mass spectrometer (SCIEX) equipped with a Duo Spray Ion Source configured for microflow HPLC applications and coupled to Eksigent ekspert nanoLC 400 System (Eksigent Technologies, Dublin, CA, USA) also configured for microflow applications. Reversed-phase chromatography used trapping for 3 min at a flow rate of 10 µL/min onto a Trajan Protecol trap (120 Å, 3 µm, 10 mm × 300 µm) followed by separation on an Eksigent ChromXP C18 3 µm 120 Å (3C18-CL-120, 3 µm, 120 Å, 0.3 × 150 mm) analytical column at a flow rate of 5 µL/min maintained at 40 °C. Trapping of peptides was done using mobile phase A (0.1% FA in water) only, whereas separation was performed on a combination of mobile phase A and B (0.1% FA in 100% ACN). A 68 min linear gradient of 3–25% mobile phase B followed by a 5 min linear gradient of 25–35% mobile phase B, was used for separation of peptides, followed by the elution of peptides. The column was flushed with 80% of Mobile phase B for 5 min after each elution, and re-equilibration was done by 97% Mobile phase A for 8 min before next samples. Inside a mass spectrometer, eluted peptides were subjected to cyclic SWATH based on an approach published by [17] with modifications according to the current experiment. Specifically, a high-resolution (30,000) TOF MS scan was collected over a range of 350–1500 *m*/*z* for 50 ms and high-sensitivity TOF MS/MS scans over a range of 100–1800 *m*/*z* over 100 variable Q1 windows (50 ms per window), resulting in the total duty cycle of 3.1 s. The collision energy for each window was set using the collision energy of a 2+ ion.

#### 2.3.4. Quantitative Analysis

Targeted data extraction of all results from plasma analysis was performed using Skyline software [34] using a designed merged dog library. Retention time calibration and the prediction were performed using spiked-in Biognosys 11 standard peptides and chromatograms were extracted within the 20 min of predicted retention time. For quantitative analysis and peptides with Cut off score—0.99, precursor charges—2,3,4; Ion charges—1,2,3; Ion types—y, b; structural modification—carbamidomethyl © with a minimum dot *p* value of 0.5 was considered for analysis. Further curation was done on the skyline document by removing repeated peptides, empty proteins, and duplicated proteins from the list. To estimate the false discovery rate (FDR) associated with target peptides, the ‘second-best peak’ scoring model was designed using the mProphet algorithm [35] in Skyline. A FDR of 1% was applied and peptides passing the q-value cut-off of 0.01 with three out of four replicates were selected for quantification. For comparative SWATH analysis, five replicates per condition were selected for quantitation. The resulted peaks were further evaluated using different quality check parameters, including dot product (dotp) values which represent a correlation between observed (library) and measured (DIA) spectra, and ‘1’ means a perfect match, mass accuracy, and coefficient of variation among technical replicates. Subsequently, to extract the report containing the list of quantified proteins and peptides and data related to SWATH extractions, the Skyline export report module was used. The settings used for targeted data extraction are submitted as Appendix A. A differential expression analysis between three different conditions, healthy, inflammatory, and miscellaneous, was performed using MSstats [36]. Proteins with absolute log2 fold change (log2 FC) > 0.5 and adjusted *p*-value < 0.05 were considered as significant. All the library visualization, comparison plots and volcano were generated using scripts written in R programming language (Version 3.4). Data was plotted using the ggplot2 R package.

## 3. Results

### 3.1. Protein Identifications from Different Digestion Techniques

Various digestion techniques were employed to build an initial dog plasma proteome repository. In the case of samples collected for healthy animals, in-solution digestion resulted in the identification of 147 protein groups; FASP digestion identified 177 protein groups and in-gel digestion identified 169 protein groups (Figure 1).

### 3.2. Protein Identifications from Different Sample Types

From the results shown above, FASP digestion was found to be superior compared to the other two digestion techniques to identify the maximum number of proteins from dog plasma. Therefore, FASP digestion was utilized for the digestion of samples collected from clinically unhealthy animals. This experiment resulted in the identification of 196 protein groups, 64 of which were uniquely identified in plasma corresponding to unhealthy conditions (Figure 2).

### 3.3. Protein Identifications after ProteoMiner

The pooled plasma samples collected from healthy dogs were processed using a ProteoMiner enrichment kit. SDS-PAGE was then used to visualise the difference between crude and treated samples (Figure 3A) and protein bands were digested for protein identification by mass spectrometry (Figure 3B). The absence of bands corresponding to albumin was evident for the ProteoMiner processed sample, which resulted in the identification of 293 protein groups. Only 169 proteins were detected in the case of crude plasma (Figure 3B).

### 3.4. Protein Identifications after ACN Precipitation

SDS-PAGE of precipitated samples (Figure 4A) demonstrated a clear difference between the protein band pattern in pellet fractions when compared with supernatant fractions. Most abundant proteins in the supernatant fraction were in the 10–75 kDa range. These were not visible and/or masked by the presence of high molecular weight proteins in the case of pellet fractions. Following in-gel digestion, a total of 211 protein groups were identified from all acetonitrile precipitated fractions (Figure 4B).

In the case of healthy dog plasma samples, ProteoMiner has resulted in the highest number of protein identifications, followed by ACN precipitation and in-gel digestion (Figure 5).

### 3.5. Protein Quantitation Using Data-Independent Acquisition

From all digestion and fractionation techniques employed, a library consisting of a total of 3119 peptides corresponding to 420 proteins (provided as the Appendix A) was generated and used for targeted extraction of SWATH data. At a 1% false discovery rate (FDR), all proteins were quantified from undepleted dog plasma samples using Skyline software that met quality and reproducibility criteria. Approximately 80% of total quantified peptides had a coefficient of variation (CV) ~25. CV plot and other parameters characterising the quantitative assay based on the SWATH approach were included in Figure 6A–E and given as Appendix A.

To assess the applicability of a developed assay to clinical samples, undepleted dog plasma samples corresponding to three different conditions: healthy, inflammatory, and miscellaneous were digested in replicates (*n* = 5) and processed using FASP digestion prior to SWATH analysis. A total of 95 (Figure 7A) and 156 (Figure 7B) proteins were significantly altered in inflammatory and miscellaneous conditions respectively when compared to healthy condition. The list of proteins specific to each condition were given as Appendix A.

## 4. Discussion

This study used different techniques to prepare samples for the canine spectral library, which was foundational for developing an assay to quantify more than 400 proteins in non-depleted plasma samples. For enabling SWATH quantitation, a reference DDA-based spectral library is required, which is genome annotated, consisting of a peptide fragmentation pattern and normalized retention time (RT) of all peptides for data extraction [37,38]. The generated reference library should contain all peptides of interest to enable detection and quantitation using SWATH [39]. Therefore, the first step in developing a spectral library is to choose a clinically relevant sample set and acquire MS/MS spectra of identified peptides [40]. The second step is to identify all possible peptides from samples, which is challenging in samples like serum/plasma due to the presence of a dynamic range of proteins [41]. Considering the stochastic nature of DDA, the generation of plasma spectral library requires depleting highly abundant proteins to reduce the complexity and increase the sensitivity of the assay [42]. In our study, we used various digestion and orthogonal fractionation techniques to increase plasma proteome coverage and the subsequent application of the developed assay to clinical samples enabled the detection of differentially abundant proteins discriminatory of the sample source which included known biomarkers of canine (Table 3) and human (Table 4) diseases. To our knowledge, this is the first time such a high number of proteins could be quantified from undepleted plasma in this species and shows the utility of the SWATH approach for clinical veterinary applications.

When generating this library, FASP digestion resulted in the identification of substantially more proteins compared to in-solution and in-gel digestion techniques (Figure 1); however, each approach contributed a unique set of proteins, thus expanding the library. The inclusion of samples corresponding to unhealthy animals and various fractionation techniques (SDS-PAGE, ProteoMiner, and ACN precipitation) further increased the number of proteins.

One concern when isolating proteins from plasma is that low abundant and low molecular weight proteins, such as tissue leakage proteins, may be obscured and the high abundant proteins need to be removed (depleted) to study low abundant proteins [61]. Several techniques, such as immuno depletion using ProteoPrep 20 and enrichment using Bio-Rad beads [62] and immune affinity depletion [63] have been used to deplete human plasma. However, each has drawbacks, including immuno-affinity kits being specific to human or laboratory rodent samples [64] and not suitable for veterinary species [65]. Fortunately, the ProteoMiner kit has been shown to be a useful depleting approach to cattle [66], pigs [65] and dogs [67]. In this study, the ProteoMiner kit has significantly (Figure 3A) depleted high abundant proteins, especially albumin in dog plasma, and had contributed a greater number of proteins (225) in this study. The ProteoMiner kit is comprised of different hexapeptide ligand beads, having an affinity to interact with all possible proteins in a sample [68]. High abundant proteins quickly oversaturate their ligands due to their abundance in plasma samples, and left-over unbound proteins will be removed in washes. On the other hand, ligand sites will be easily available for enriching low abundant proteins [69]. The results from previous studies and this current study proves that the ProteoMiner enrichment kit could be used for depleting high abundant proteins from plasma/serum in veterinary species.

Another significant advance in the current study was the fractionation of dog plasma using an organic solvent. Acetonitrile precipitation of plasma successfully depleted high abundant proteins in human serum [70], including when used under a range of pH conditions (5.0, 7.0, and 9.0) [32]. Acetonitrile precipitation of dog plasma at different pH (3.5, 5, 8 & 9) does not completely remove the high abundant proteins like the ProteoMiner kit, but efficiently increased the identification of a greater number of proteins. This may be due to the effect of pH on protein solubility and molecular weight [70] and, similar to what has been reported for human serum samples [71], the use of the ProteoMiner enrichment kit followed by acetonitrile precipitation was superior for depleting high abundant proteins.

The quality and reproducibility of SWATH extractions were investigated using pooled healthy and unhealthy plasma samples as technical replicates (*n* = 4). Different parameters were applied to examine the reproducibility of the developed assay. As shown in Figure 6B, the majority of quantified peptides have coefficient of variation (CV) values less than 25% among technical replicates, such reproducibility in quantified values can also be visualized in Figure 6E, which shows the overall consistent intensity range within each sample or technical replicate. Moreover, in order to keep the high quality peptides, a q-value cut-off of 0.01 (1% FDR) was applied. Figure 6C shows the distribution of q-values for the quantified peptides, where virtually all of the peptides have a q-value ≤ 0.01. Mass error shows the measure of difference between actual and theoretical mass of an ion and should be close to 0 ppm for a high quality spectra. Figure 6D shows the distribution of mass error values with the mean value of 0 ppm and an overall range of −10 to 10 ppm. These parameters add confidence to the protein and peptide quantifications among different samples and within replicates of the samples in the experiment. In this study, a spectral library was generated using information (in regards to instrumentation, and software analysis) from previously published DDA libraries in humans [14] and model animals [19,20,72]. Detailed information on generating spectral libraries, data acquisition, and DIA analysis has been discussed elsewhere [73,74]. In dogs, very few studies have reported spectral libraries using saliva [75] and tears [76]. None of these studies have evaluated the application of those libraries to deconvolute SWATH data. Recently, SWATH was used to explore serum proteomic alterations in different stages of *Leishmania* infected dogs [24]. In that study, a local spectral library was generated consisting of only 169 proteins (using SDS-PAGE fractionation), which enabled the quantitation of only 44 significantly changed proteins specific to one disease condition in dogs. In contrast, our study had identified and quantitated the highest number of plasma proteins, which proves that the composition of the spectral library determines the efficiency of SWATH quantitation of proteins from undepleted plasma [77]. The advantage of our study was to include various clinical samples subjected a varied clinical conditions, which improved the quality of this plasma spectral library [78], and could be readily used to explore the pathophysiology of various clinical conditions. The main objective of this study was to develop a reproducible method that can quantify a maximal number of proteins in healthy and unhealthy subjects from undepleted plasma samples, which we believe was achieved in this study (Figure 6 and Figure 7). The next step in our workflow is to apply this spectral library to investigate specific diseases in dogs. Recently, studies published from our research team have shown the importance of DDA libraries and their applicability to identify plasma biomarkers in cattle [25], and also discussed how combining spectral libraries could enable identification and quantitation of proteins [26] in similar species using SWATH.

It should also be noted that considering recent technical advances in the field of quantitative proteomics in human medicine, SWATH analysis could now be performed using in silico libraries using DeepDIA [79], and Prosit [80] instead of developing in-site DDA libraries. However, this could be possible only due to the availability of a fully annotated genome, which is currently lacking in all veterinary species [8]. This could be the next step to consider for biomarker discovery in veterinary proteomics.

## 5. Conclusions

The current study has shown that a highly multiplexed and reproducible assay capable of quantifying > 400 proteins in non-depleted plasma can be established using a purpose-built spectral library. The proteins identified included acute phase proteins, oxidative stress markers and many other plasma proteins important to diagnosing canine diseases, such as osteoarthritis, pyometra and canine leishmaniosis. The assay and the published protocols can be utilized for biomarker discovery, diagnosis, and prognosis, while the methods and pipeline used may be useful for generating spectral libraries and establishing quantitative assays in other veterinary species.

## Figures and Tables

**Figure 1 proteomes-10-00009-f001:**
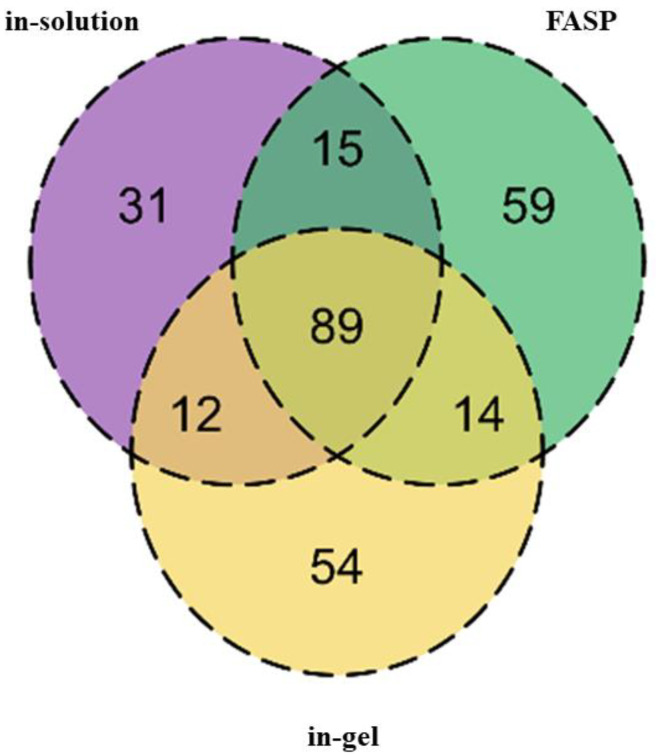
The Venn diagram represents the total number of identified proteins from plasma of healthy dogs after processing using different digestion techniques. Each digestion technique enabled the identification of different protein groups.

**Figure 2 proteomes-10-00009-f002:**
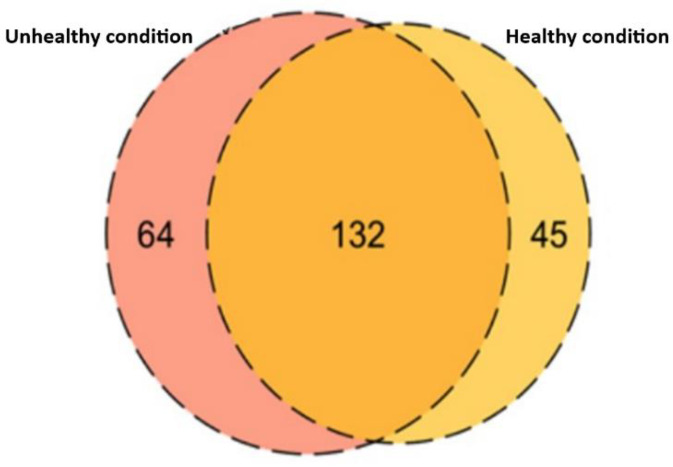
The Venn diagram represents the comparison of a total number of proteins identified in plasma collected from healthy and unhealthy animals. A total of 132 proteins were overlapped between the conditions and 64 proteins were unique to unhealthy and 45 proteins were unique to healthy conditions.

**Figure 3 proteomes-10-00009-f003:**
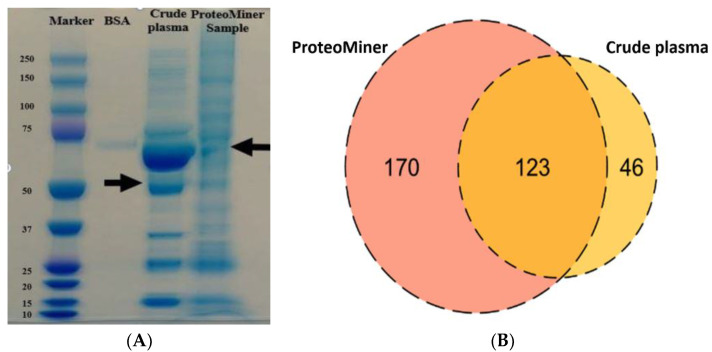
(**A**) SDS-PAGE of crude and ProteoMiner processed samples. Lane 1 = MW marker; Lane 2 = BSA; Lane 3 = crude dog plasma sample and Lane 4 = ProteoMiner treated healthy dog plasma sample. The arrows indicate the main difference in the protein band patterns and corresponding to depletion of albumin in the case of the treated sample. (**B**) Venn diagram represents the comparison of the total number of proteins identified in crude dog plasma and ProteoMiner processed plasma collected from healthy animals. ProteoMiner treatment enabled identification of the highest number of unique proteins (170) over a crude untreated plasma sample (46 unique proteins).

**Figure 4 proteomes-10-00009-f004:**
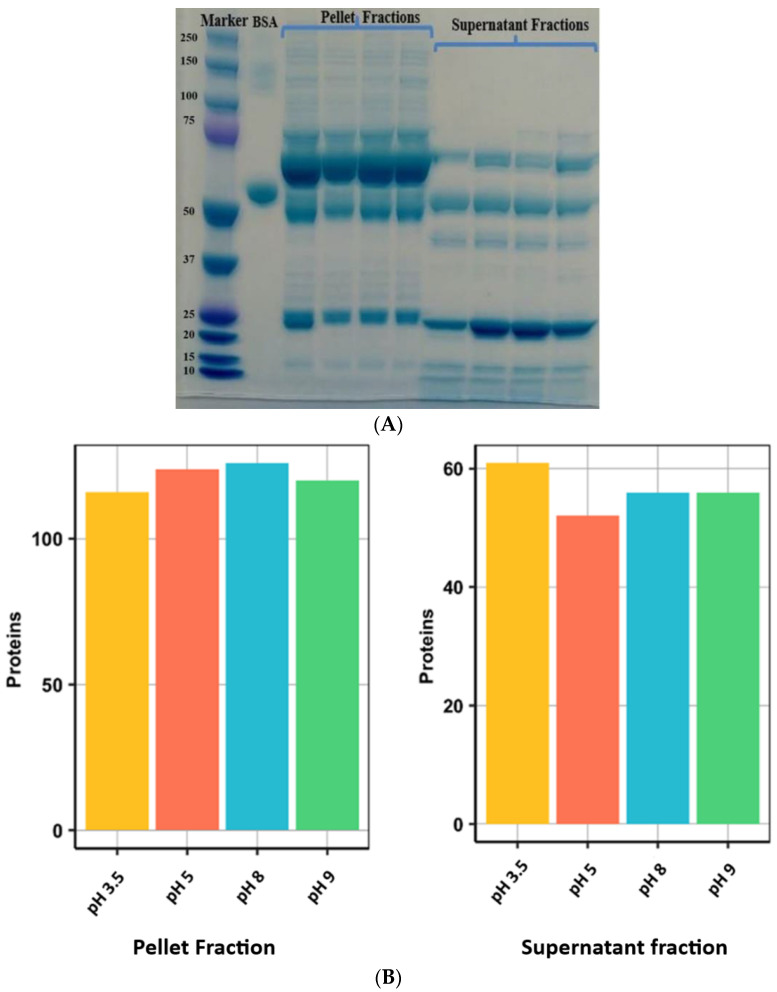
(**A**) SDS-PAGE of supernatant and pellet fractions of ACN precipitated healthy dog plasma samples. Lane 1 = MW marker; Lane 2 = BSA; Lanes 3–6 = pellet fractions collected at pH 3.5, 5.0, 8.0 and 9.0, respectively; Lanes 7–10 = supernatant fractions collected at pH 3.5, 5.0, 8.0 and 9.0, respectively. (**B**) Bar diagrams showing the total number of identified proteins from plasma of healthy dogs in different fractions resulting from ACN precipitation and collected at different pH. Each bar represents an absolute number of protein groups identified in various fractions. At different pH, protein precipitation resulted in the identification of a similar number of proteins.

**Figure 5 proteomes-10-00009-f005:**
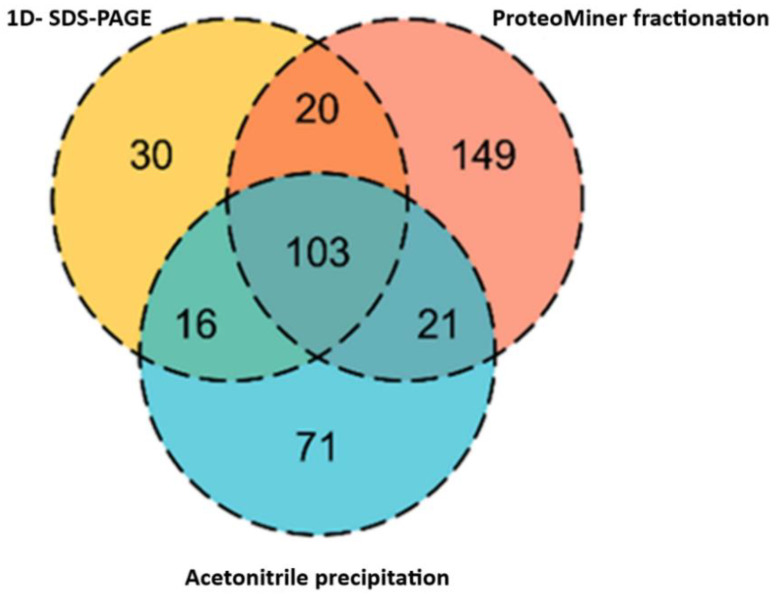
Venn diagram showing the contribution of different fractionation techniques to proteins represented in a spectral library. Most of the proteins were overlapped between the techniques, however, the highest number of unique proteins (149) were identified in ProteoMiner fraction, followed by ACN precipitation (71 unique proteins) and least identified in SDS-PAGE (30 unique proteins).

**Figure 6 proteomes-10-00009-f006:**
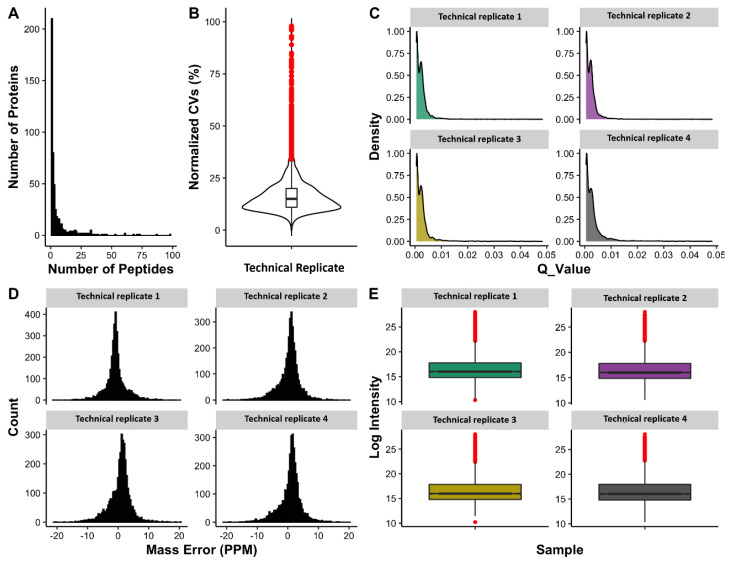
Characteristics of an assay to quantify 400 proteins in dog plasma. (**A**) Frequency plot of the number of peptides quantified per protein. (**B**) Violin plot of coefficient of variation (CV%) among technical replicates—confirming the consistency of peptide quantification among technical replicates. (**C**) Distribution of q-values for extracted peaks. (**D**) Distribution of mass errors for extracted product ions. (**E**) Logged peptides intensity from all technical replicates showing the reproducibility of peptide quantitation.

**Figure 7 proteomes-10-00009-f007:**
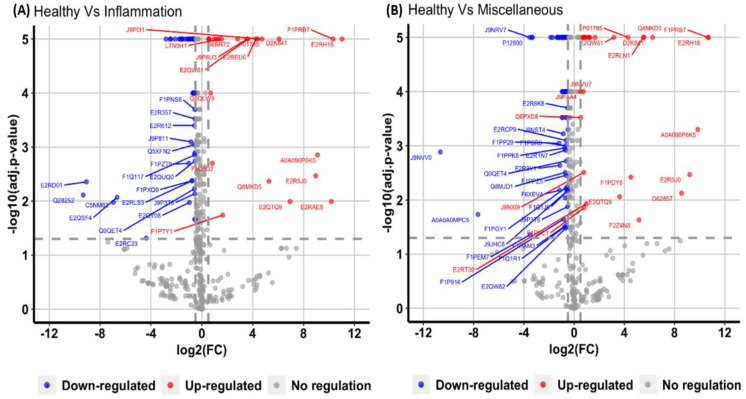
Volcano plot demonstrating changes in proteins corresponding to (**A**) inflammatory condition and (**B**) miscellaneous disease conditions, when compared to healthy samples. The X-axis displays the magnitude of fold changes and the Y-axis shows the statistical significance. The horizontal line indicates the adjusted *p*-value cut-off of <0.05, and vertical lines denote the absolute log 2-fold change cut off of 0.5. The blue dots represent downregulated and the red dots represent upregulated proteins in different clinical conditions.

**Table 1 proteomes-10-00009-t001:** Samples used for generating peptide spectral library specific to dog blood plasma.

No.	Condition	No. of Animals Used to Generate a Pooled Sample	Digestion Technique	Fractionation/Enrichment Technique
1	Healthy	8	In-solution Filter-aided In-gel	1D—SDS PAGE ProteoMiner enrichment Acetonitrile precipitation
2	Unhealthy (inflammatory and miscellaneous conditions)	26	Filter-aided	None

**Table 2 proteomes-10-00009-t002:** Samples used for quantitative profiling of dog blood plasma proteins.

No.	Condition	No. of Animals Used to Generate a Pooled Sample	Digestion Technique	Fractionation/Enrichment Technique
1	Healthy	8	Filter-aided	None
2	Inflammatory conditions	8	Filter-aided	None
3	Miscellaneous conditions	18	Filter-aided	None

**Table 3 proteomes-10-00009-t003:** List of proteins present in the current generated spectral library and their involvement in different diseases of dogs.

Sl. No.	Disease Condition	Proteins Studied	Reference
1	Canine Babesiosis	Alpha 1 acid glycoprotein, Apolipoprotein A-1, Complement c3, Hemopexin, Alpha 2-HS glycoprotein, Haptoglobin, Clusterin	[43]
2	Canine lymphoma	Apolipoprotein A-I, Apolipoprotein C-I, Apolipoprotein C-II, Apolipoprotein C-III, Apolipoprotein E, Beta-2-glycoprotein 1, Clusterin, Coagulation factor IX, Fibrinogen alpha chain, Fibrinogen beta chain (Fragment), Fibrinogen gamma chain Fibronectin, Haptoglobin, Plasminogen Serum amyloid A protein, Transferrin receptor protein 1	[44]
4	Canine Pyometra	Alpha-1-acid glycoprotein 1, Haptoglobin, Alpha-2-macroglobulin, Hemopexin, Transthyretin, Transferrin receptor protein, Retinol-binding protein, Gelsolin, Alpha 2-HS glycoprotein	[45]
5	Canine Mammary tumors	Alpha-1-microglobulin/bikunin precursor, Angiotensinogen, Serum albumin, Gelsolin	[46]
6	Canine Chronic Valve disease	Apolipoprotein B, Apolipoprotein M, Apolipoprotein D	[47]
7	H3N2 canine Influenza virus	Haptoglobin, Apolipoprotein E, Alpha 1 acid glycoprotein, Beta-2-microglobulin	[48]
8	Duchenne muscular dystrophy	Alpha-1-B glycoprotein, Alpha 2-HS glycoprotein, Fetuin B, Hemopexin, Tropomyosin 2	[49]
10	Canine Encephalitis	Hemopexin, Gelsolin, Transthyretin, Beta-2-glycoprotein 1 Apolipoprotein E	[50]
11	Canine Leishmaniasis	Haptoglobin, Hepatocyte growth factor activator, Hyaluronan binding protein 2 Sulfhydryl oxidase, Complement C8 alpha chain, Complement C9	[24]

**Table 4 proteomes-10-00009-t004:** List of proteins identified as biomarkers of different human diseases present in current dog spectral library.

Sl. No.	Protein Name	Human Disease Condition	Reference
1	Gelsolin	Glioma	[51]
2	Ceruloplasmin	Identified as a protein biomarker in prostate cancer	[52]
3	Haptoglobin	Identified as biomarker in lung adeno carcinoma	[53]
4	Alpha 1 antitrypsin	Studied in human breast cancer	[54]
5	Vimentin	Expressed in human and canine osteosarcoma cells	[55]
6	Tropomyosin 3	Upregulated in metastatic carcinomas	[56]
7 8	Myosin light chain 2 Tropomyosin 1	Downregulated in metastatic mammary carcinoma, which is also expressed in human breast cancer	
9	Triosephosphate isomerase	Analysed as autoantigens from the canine mammary cell line	[57]
10 11	Transthyretin Apolipoprotein A-I	Identified as biomarkers for detecting early stage of ovarian cancer in human	[58]
12	Tissue inhibitor of metalloproteinase 1 (Fragment)	Pancreatic cancer	[59]
13	Alpha 2-HS glycoprotein Transthyretin	Colorectal cancer	[60]

## Data Availability

The data presented in this study are available on request from the corresponding author.

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
