# Peer review of "Data-Independent Acquisition Enables Robust Quantification of 400 Proteins in Non-Depleted Canine Plasma"

_proteomes, 2022, doi:10.3390/proteomes10010009_

Round 1
Reviewer 1 Report
The paper entitled “Data-Independent Acquisition Enables Robust Quantification 2 of 400 Proteins in Non-Depleted Canine Plasma” is well structured and presented. The main topics of the paper are well introduced and the results and discussion are well organized.
The proposed approrach is very relevant for quantitative assays in dogs and contribute significantly for the potential improvemnt in clinical veterinary applications for diagnosis and new biomarkers identification.
Reviewer 2 Report
This is very interesting manuscript. The presentation of results and their discussions are good. I have some very minor comments-
- Central message in the abstract is missing.
- Many recent findings are missing in the introduction section to support the present study.
- Authors may define and expand the figure captions.
- Separate last paragraph of the discussion section with conclusion.
Reviewer 3 Report
Authors have used DDA mass spectrometry for quantitative and qualitative analysis of canine plasma proteins. They were able to identify 400 proteins collected from healthy and unhealthy dogs. The work is interesting, and the protein library generated by them can prove to be highly beneficial for researchers working in the field. However, I have serious concerns regarding the way the manuscript is written. There is a lack of discussions and details which makes it difficult to understand. If the authors take care of them, I recommend the article to be accepted for publication in Proteomes.
The title says “Data-Independent Acquisition Enables…” whereas page 2, line 55 mentions “This study utilized DDA mass spectroscopy…..”. Please rewrite in such a way that it is easily understandable without any confusion.
The title focuses on the use of DIA so a little more discussion on DIA will be appreciated. Some easy-to-understand sentences on the following points will be helpful:
How is it different from conventional techniques?
Is there any m/z range where the conventional one fails but DIA works well?
Is there a similar work where conventional mass spec is used to generate a library of proteins? If so, please discuss it in one or two sentences and show how your work is significantly better.
Is there any difference in sample requirements for conventional and DIA?
What is the major difference between DIA and DDA?
Just by reading the manuscript it is difficult to understand why both DDA and DIA was performed. What specific information they provide? The result and discussion section doesn’t say anything about it which I consider as one of the most serious concerns.
One of the key points of the manuscripts is the use of ProteoMiner kit. I see a detailed description of its usefulness in the description section. Considering that without the use of this kit the library generation was not possible, I would suggest including the term “protein Enrichment” in the title and in the keywords. Also, as the “materials and methods” section comes first, consider including one line regarding the purpose of the kit before explaining its usage (section 2.2.4).
Section 2.3.1., line 175: “Peptides with intensity greater than 150 cps and charge state of 2-5, were selected for the analysis.” How was this threshold decided?
In section 3.2, line 255: “Given FASP digestion was superior….” It is not clear to me whether it already established or the authors are concluding this from the results of the healthy samples? If if is already established then what was the need for trying other two techniques?
There is a lack of discussion on Figure 6 and Figure 7 in the main text. Please include a good explanation of the observation and any conclusion drawn from it.
Did the authors try to segregate the proteins based on the specific disease condition and/or gender/age of the source?
Is it possible to include the details of all the proteins as a table (like the details of the altered proteins are given as tables S4 and S5)?
